# Teaching a GAN What Not to Learn

**Siddarth Asokan**\*
Robert Bosch Center for Cyber-Physical Systems
Indian Institute of Science
Bangalore, India
siddartha@iisc.ac.in

**Chandra Sekhar Seelamantula**
Department of Electrical Engineering
Indian Institute of Science
Bangalore, India
css@iisc.ac.in

## Abstract

Generative adversarial networks (GANs) were originally envisioned as unsupervised generative models that learn to follow a target distribution. Variants such as conditional GANs, auxiliary-classifier GANs (ACGANs) project GANs on to supervised and semi-supervised learning frameworks by providing labelled data and using multi-class discriminators. In this paper, we approach the supervised GAN problem from a different perspective, one that is motivated by the philosophy of the famous Persian poet Rumi who said, *"The art of knowing is knowing what to ignore."* In the GAN framework, we not only provide the GAN *positive* data that it must learn to model, but also present it with so-called *negative* samples that it must learn to avoid — we call this *the Rumi framework*. This formulation allows the discriminator to represent the underlying target distribution better by learning to penalize generated samples that are undesirable — we show that this capability accelerates the learning process of the generator. We present a reformulation of the standard GAN (SGAN) and least-squares GAN (LSGAN) within the Rumi setting. The advantage of the reformulation is demonstrated by means of experiments conducted on MNIST, Fashion MNIST, CelebA, and CIFAR-10 datasets. Finally, we consider an application of the proposed formulation to address the important problem of learning an under-represented class in an unbalanced dataset. The Rumi approach results in substantially lower FID scores than the standard GAN frameworks while possessing better generalization capability.

## 1 Introduction and Related Work

Generative adversarial networks (GANs), originally proposed by Goodfellow *et al.* [1], are unsupervised deep learning machines, wherein a generator learns to mimic a target data distribution, whereas the discriminator attempts to distinguish between real data and the samples coming from the generator. The original GAN optimization and subsequent flavors such as the least-squares GAN (LSGAN) [2] and $f$-GANs could be viewed as performing divergence minimization, where the optimal generator is the minimizer of a chosen divergence metric between the generated and true data distributions.

The generator in the standard GAN formulation transforms input noise $z \sim p_Z$, typically a standard multivariate Gaussian, to the output $G(z)$ with distribution $p_g(x)$. The target data is sampled from an underlying distribution $p_d(x)$. The discriminator $D(x)$ predicts the probability of its input coming from $p_d$. This is formulated as a min-max game between the generator and the discriminator:

$$\min_{p_g} \max_{D(x)} \quad \mathbb{E}_{x \sim p_d}\left[\log D(x)\right] + \mathbb{E}_{x \sim p_g}\left[\log(1 - D(x))\right],$$

where the optimal discriminator $D^*(x) = \frac{p_d}{p_d + p_g}$ was shown to be the one that measures the odds of a sample coming from the data distribution, and the optimal generator was shown to be the minimizer of

---

the Jensen-Shannon divergence between $p_d$ and $p_g$. In the LSGAN formulation [2], the discriminator executes a regression task, i.e., to assign a class label $a$ to the generated samples, and a class label $b$ to the real ones. The generator learns to confuse the discriminator by creating samples that get classified by the discriminator as belonging to another category labelled $c$. Mathematically stated, the LSGAN optimization comprises the following problems:

$$\min_{D(\boldsymbol{x})} \quad \mathbb{E}_{\boldsymbol{x} \sim p_d}\left[(D(\boldsymbol{x}) - b)^2\right] + \mathbb{E}_{\boldsymbol{x} \sim p_g}\left[(D(\boldsymbol{x}) - a)^2\right], \text{ and}$$

$$\min_{p_g} \quad \mathbb{E}_{\boldsymbol{x} \sim p_d}\left[(D(\boldsymbol{x}) - c)^2\right] + \mathbb{E}_{\boldsymbol{x} \sim p_g}\left[(D(\boldsymbol{x}) - c)^2\right].$$

The optimal generator in this case is the minimizer of the Pearson-$\chi^2$ divergence between $2p_g$ and the sum of $p_d$ and $p_g$.

Conditional GANs (CGANs) [3] are a supervised extension of GANs and require labelled input data. The objective in CGANs is to control the class of the generated sample. Conditional GANs augment the inputs to both the generator and the discriminator with a one-hot encoding of the class label. Subsequent works have improved upon the class-conditional image generation in CGANs by introducing the label information into the hidden layers of the discriminator [4] or by means of an inner-product with the class label introduced into the pre-final layer of the discriminator (CGAN-PD) [5]. Conditional GANs and their variants have found applications in text-to-image synthesis [4, 6] and image-to-image translation tasks [7–10]. They have also been modified to include multi-class discriminators [11], and perform auxiliary classification (ACGAN) [12] to improve the performance of the conditional generator. Twin auxiliary classifier GANs (TACGANs) [13] improve the multi-class performance of ACGANs by introducing a data balancing term in the generator loss. Balancing GANs (BAGANs) [14] perform data balancing by deploying an autoencoder to initialize the ACGANs. Both CGANs and ACGANs have been successful in medical image data augmentation applications [15, 16], where a converged generator is used to output samples from an under-represented class.

Another line of work involves splitting data into positive and negative samples for discriminative learning. In metric learning [17–20] and representation learning [21, 22], the relative distances between samples are used to train a neural network. The contrastive loss [17] compares pairs of samples, and assigns positive weights to similar/desired pairs and negative weights to dissimilar/undesired ones. As an extension, in triplet loss [18], the distance of the target from the positive class is minimized, and that from the negative class is maximized.

In complementary CGAN (CCGAN) [23], a CGAN is trained on data where each sample is associated with a *complementary label*, which is a yes/no tag, corresponding to a randomly selected class. Generative positive-unlabelled learning (GenPU) [24] involves training semi-supervised GANs with a mixture of positive, negative, and unlabelled samples, with the goal of obtaining a classifier that separates the positive unlabelled samples from the negative ones. Unlike the standard GAN, where the optimal generator is of ultimate interest, in CCGAN and GenPU, the end product is the optimized discriminator.

## 2 Our Contribution

The approach that we advocate in this paper is motivated by the famous Persian poet Rumi's philosophy, *"The art of knowing is knowing what to ignore."* In the context of machine learning, we interpret it as empowering models to *learn by ignoring*. Formally, this represents learning both from examples as well as counterexamples. We refer to our formulation as *The Rumi Formulation*. We take the "middle-ground" between the conditional data generation capability of ACGANs, and the positive, unlabelled, and negative data classification feature of models such as GenPU. Figure 1 brings out the differences between these approaches and the proposed one.

In Rumi-GAN, the discriminator learns to bin the samples it receives into one of three classes: (1) Positives, representing samples from the target distribution; (2) Negatives, representing samples from the same underlying background distribution as that of the target data, but from a subset that must be avoided; and (3) Fakes, which are the samples drawn from the generator. The generator, on the other hand, is tasked with learning the distribution of only the positive ones by simultaneously learning to avoid the negative ones. Effectively, the Rumi-GAN philosophy is a unique case of *complementary*

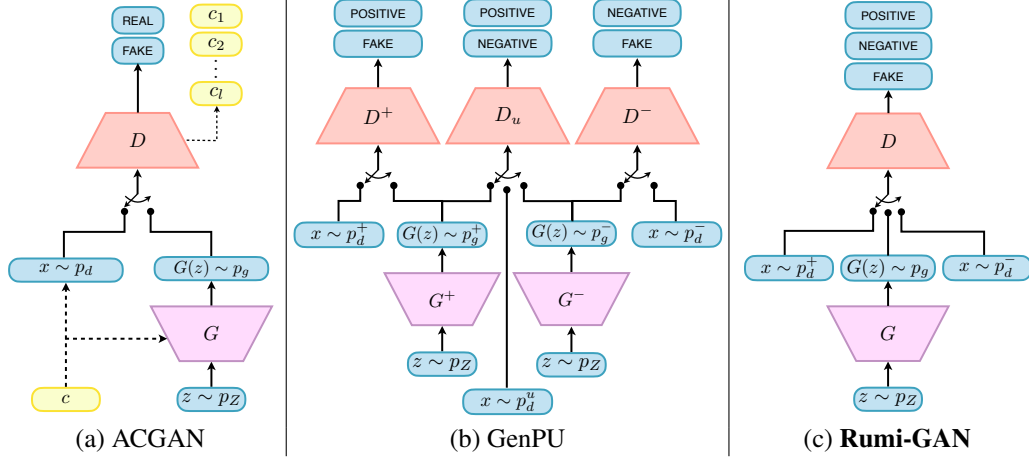

Figure 1: (🎨 Color Online) Comparison of the pipelines of ACGAN and GenPU vis-à-vis the proposed Rumi-GAN.

*learning*, where the negative class of exemplars serve the purpose of boosting the positive-class learning capability of the generator.

This paper is structured as follows. In Section 3, we first reformulate the standard GAN (SGAN) and LSGAN within the Rumi framework, giving rise to Rumi-SGAN and Rumi-LSGAN, respectively, and derive the optimal distribution learnt in each case. Any other flavor of GAN could also be accommodated within the Rumi framework. In Section 4, we demonstrate the learning capabilities of Rumi-LSGAN on MNIST and Fashion-MNIST datasets and show improvements over the baseline LSGAN and ACGAN approaches. In Section 5, considering the particular case of training on unbalanced datasets, we simulate minority classes in MNIST, CelebA and CIFAR-10 datasets, and show that Rumi-GAN learns a better representation of the target distribution and has a lower generator network complexity when compared with the state-of-the-art ACGAN and CGAN architectures. Although we demonstrate the applicability of Rumi-GANs on unbalanced datasets, the formulation is extendable to any complementary learning task.

## 3 The Rumi Formulation of GANs

Here, we consider a scenario where the real data distribution consists of either overlapping or non-overlapping subsets, one labeled positive and the other negative. In the Rumi formulation of SGAN and LSGAN, the discriminator is converted into a three-class classifier giving scalar outputs. The data distribution $p_d$ is split into two: (i) the target distribution that the GAN is required to learn ($p_d^+$); and (ii) the distribution of samples that it must avoid ($p_d^-$). Unlike ACGANs, in the Rumi formulation, we weigh the generator samples from the desired and undesired classes differently. In principle, one could develop the Rumi counterpart of any known $f$-divergence [25] or integral probability metric based GAN [26, 27]. The details are deferred to the supplementary document. In the following, we present the optimal discriminator and generator functions obtained for Rumi-SGAN and Rumi-LSGAN.

### 3.1 The Rumi-SGAN

The Rumi-SGAN discriminator loss comprises three terms: the expected cross-entropy between (a) $[1,0]^{\mathrm{T}}$ and $[D(\boldsymbol{x}), 1-D(\boldsymbol{x})]^{\mathrm{T}}$ for the positive data; (b) $[0,1]^{\mathrm{T}}$ and $[D(\boldsymbol{x}), 1-D(\boldsymbol{x})]^{\mathrm{T}}$ for the generator samples; and (c) $[0,1]^{\mathrm{T}}$ and $[D(\boldsymbol{x}), 1-D(\boldsymbol{x})]^{\mathrm{T}}$ for samples drawn from the negative class, and is given as:

$$\mathcal{L}_D^S = -\left(\alpha^+ \, \mathbb{E}_{\boldsymbol{x}\sim p_d^+}\left[\log D(\boldsymbol{x})\right] + \mathbb{E}_{\boldsymbol{x}\sim p_g}\left[\log(1-D(\boldsymbol{x}))\right] + \alpha^- \, \mathbb{E}_{\boldsymbol{x}\sim p_d^-}\left[\log(1-D(\boldsymbol{x}))\right]\right), \quad (1)$$

where $\alpha^+$ and $\alpha^-$ are the weights attached to the losses associated with the positive and negative subsets, respectively. The min-max generator considers $\mathcal{L}_G = -\mathcal{L}_D$, subject to the integral constraint

$\Omega_{p_g} : \int_{\mathcal{X} \subseteq \mathbb{R}^n} p_g(\boldsymbol{x}) \, \mathrm{d}\boldsymbol{x} = 1$, and the non-negativity constraint $\Phi_{p_g} : p_g(\boldsymbol{x}) \geq 0, \ \forall \, \boldsymbol{x}$. Incorporating the constraints, we consider the Lagrangian

$$\mathcal{L}_G^S(D^*(\boldsymbol{x})) = -\mathcal{L}_D^S(D^*(\boldsymbol{x})) + \lambda_p \left( \int_{\mathcal{X}} p_g(\boldsymbol{x}) \, \mathrm{d}\boldsymbol{x} - 1 \right) + \int_{\mathcal{X}} \mu_p(\boldsymbol{x}) p_g(\boldsymbol{x}) \, \mathrm{d}\boldsymbol{x}, \qquad (2)$$

where $\lambda_p$ and $\mu_p(\boldsymbol{x})$ are the Karush-Kuhn-Tucker (KKT) multipliers with $\mu_p(\boldsymbol{x}) \leq 0, \ \forall \, \boldsymbol{x} \in \mathcal{X}$, and $\mu_p(\boldsymbol{x}) p_g^*(\boldsymbol{x}) = 0, \ \forall \, \boldsymbol{x} \in \mathcal{X}$. The optimal KKT multipliers will have to be determined.

**Lemma 3.1.** *The optimal Rumi-SGAN: Consider the GAN optimization defined through Equations* (1) *and* (2). *The optimal discriminator $D^*(\boldsymbol{x})$ and the optimal generator density $p_g^*(\boldsymbol{x})$ are given as*

$$D^*(\boldsymbol{x}) = \frac{\alpha^+ p_d^+(\boldsymbol{x})}{\alpha^+ p_d^+(\boldsymbol{x}) + p_g(\boldsymbol{x}) + \alpha^- p_d^-(\boldsymbol{x})} \quad and \quad p_g^*(\boldsymbol{x}) = (1 + \alpha^-) p_d^+(\boldsymbol{x}) - \alpha^- p_d^-(\boldsymbol{x}),$$

*respectively, where $\alpha^- \geq \alpha^+ - 1$, and $\alpha^+ \in [0, 1]$. The optimal KKT multipliers are $\mu_p^*(\boldsymbol{x}) :=$ $0, \ \forall \, \boldsymbol{x} \in \mathcal{X}$, and $\lambda^* = \log\left(\frac{1 + \alpha^+ + \alpha^-}{1 + \alpha^-}\right)$.*

**Proof**: The cost functions in Equations (1) and (2) could be minimized point-wise. The solution to (1) is relatively straightforward since there are no constraints. Minimization of (2) subject to $\Omega_{p_g}$ and $\Phi_{p_g}$ gives $p_g^* = \left(\frac{\kappa(\boldsymbol{x})}{1 - \kappa(\boldsymbol{x})}\right) \alpha^+ p_d^+(\boldsymbol{x}) - \alpha^- p_d^-(\boldsymbol{x})$, where $\kappa(\boldsymbol{x}) = e^{-\lambda_p - \mu_p(\boldsymbol{x})}$. Enforcing the integral and non-negativity constraints, $\Omega_{p_g}$ and $\Phi_{p_g}$, respectively, yields the optimal solution. $\qquad \square$

## 3.2 The Rumi-LSGAN

The Rumi-LSGAN loss minimizes the least-squares distance between the discriminator output $D(\boldsymbol{x})$ and: (a) the class label $b^+$ for positive samples; (b) the class label $b^-$ for negative samples; and (c) the class label $a$ for samples coming from the generator. Simultaneously, the generator minimizes the least-squares distance between $D(\boldsymbol{x})$ and a class label $c$ for all the samples it generates. The weighted loss functions become

$$\mathcal{L}_D^{LS} = \beta^+ \, \mathbb{E}_{\boldsymbol{x} \sim p_d^+} \left[ \left( D(\boldsymbol{x}) - b^+ \right)^2 \right] + \beta^- \, \mathbb{E}_{\boldsymbol{x} \sim p_d^-} \left[ \left( D(\boldsymbol{x}) - b^- \right)^2 \right] + \mathbb{E}_{\boldsymbol{x} \sim p_g} \left[ \left( D(\boldsymbol{x}) - a \right)^2 \right],$$
$$(3)$$

$$\mathcal{L}_G^{LS} = \beta^+ \, \mathbb{E}_{\boldsymbol{x} \sim p_d^+} \left[ \left( D^*(\boldsymbol{x}) - c \right)^2 \right] + \beta^- \, \mathbb{E}_{\boldsymbol{x} \sim p_d^-} \left[ \left( D^*(\boldsymbol{x}) - c \right)^2 \right] + \mathbb{E}_{\boldsymbol{x} \sim p_g} \left[ \left( D^*(\boldsymbol{x}) - c \right)^2 \right], \quad (4)$$

where $\mathcal{L}_G^{LS}$ is subjected to the integral and non-negativity constraints $\Omega_{p_g}$ and $\Phi_{p_g}$, respectively.

**Lemma 3.2.** *The optimal Rumi-LSGAN: Consider the LSGAN optimization problem defined through Equations* (3) *and* (4). *Assume that the labels satisfy $a \leq \frac{b^+ + b^-}{2}$ with $b^+ > b^-$. The optimal discriminator and generator are given as*

$$D^*(\boldsymbol{x}) = \frac{b^+ \beta^+ p_d^+ + b^- \beta^- p_d^- + a p_g}{\beta^+ p_d^+ + \beta^- p_d^- + p_g} \quad and \quad p_g^*(\boldsymbol{x}) = \beta^+ \eta^+ p_d^+(\boldsymbol{x}) + \beta^- \eta^- p_d^-(\boldsymbol{x}), \quad (5)$$

*respectively, where $\eta^+ = \left(\frac{(1 + \beta^-)(a - b^+) - \beta^-(a - b^-)}{\beta^+(a - b^+) + \beta^-(a - b^-)}\right)$, and $\eta^- = \left(\frac{(1 + \beta^+)(a - b^-) - \beta^+(a - b^+)}{\beta^+(a - b^+) + \beta^-(a - b^-)}\right)$.*

**Proof**: As in the case of SGANs, the optimization of the costs in Equations (3) and (4), with $\mathcal{L}_G^{LS}$ subjected to the constraints $\Omega_{p_g}$ and $\Phi_{p_g}$, is carried out point-wise. The detailed proof is presented in the supplementary document. $\qquad \square$

## 3.3 The Optimal Generator

Let us now analyze the optimal generator distributions obtained in Lemmas 3.1 and 3.2. The Rumi-SGAN generator $p_g^* = (1 + \alpha^-) p_d^+ - \alpha^- p_d^-$ subject to $\alpha^- \geq \alpha^+ - 1$, and $\alpha^+ \in [0, 1]$ has a family of solutions based on the choice of the weights $\alpha^-$ and $\alpha^+$. The generator always learns a mixture of $p_d^+$ and $p_d^-$, and it latches on to $p_d^+$ when $\alpha^- = 0$, or $p_d^-$ when $\alpha^+ = 0$. These corner cases represent the scenarios where the corresponding loss-terms are ignored and the Rumi-SGAN coincides with the standard GAN formulation.

The optimal Rumi-LSGAN generator in Equation (5) shows that the weights and the class labels can be leveraged to control the learnt mixture density. In particular, when $\beta^+ = \beta^-$, the separation between $a, b^+$, and $b^-$ decides the mixture weights: when the negative class label $b^-$ is closer to $a$, the generator gives prominence to $p_d^+$ and vice versa. On the other hand, when $a = \frac{b^+ + b^-}{2}$, $\beta^+$ and $\beta^-$ control the mixing. As a special case, when $a < \frac{b^+ + b^-}{2}$, and $\beta^+ = \frac{a - b^-}{b^- - b^+}$, we have the optimal generator $p_g^* = p_d^+$. This is important as we can now train a discriminator using samples taken from both the positive and the negative classes while still learning the distribution of the positive class only.

## 4   Experimental Validation

We conduct experiments on MNIST [28], Fashion-MNIST [29], CelebA [30] and CIFAR-10 [31] datasets. The GAN models are coded in TensorFlow 2.0 [32]. The generator and discriminator architectures are based on deep convolutional GAN [33]. In all the cases, latent noise is drawn from a 100-dimensional standard Gaussian $\mathcal{N}(\mathbf{0}_{100}, \mathbb{I}_{100})$. The ADAM optimizer [34] with learning rate $\eta = 10^{-4}$ and exponential decay parameters for the first and second moments $\beta_1 = 0.50$ and $\beta_2 = 0.999$ is used for training both the generator and the discriminator. A batch size of 100 is used for all the experiments and all models were trained for 100 epochs, unless stated otherwise. For every step of the Rumi-LSGAN generator, the discriminator performs two updates, one with the positive data and another one with the negative data. In order to make a fair comparison, for every update of the discriminator and generator of Rumi-LSGAN, we perform two updates of the discriminator and one update of the generator on the baseline approaches. This approach of updating the discriminator multiple times per update of the generator is actually in favor of the baseline GANs as shown in [26, 35]. As argued in Section 3.3, while the Rumi-SGAN learns a mixture of the positive and negative data distributions, the Rumi-LSGAN is capable of latching on to the positive data distribution only. In view of this property, we report comparisons between Rumi-LSGAN and the baseline LSGAN [2] and ACGAN [12]. We set $(b^-, a, b^+, c) = (-1, 0, 2, 1.5)$ for Rumi-LSGAN, and use $(a, b, c) = (0, 1, 1)$ for the baseline LSGAN as proposed by Mao *et al.* [2]. Given the batches of positive, negative, and generated samples: $\mathcal{D}^+ = \{\boldsymbol{x}_i; \ \boldsymbol{x}_i \sim p_d^+, \ i = 1, 2, \ldots, N\}$, $\mathcal{D}^- = \{\boldsymbol{x}_j; \ \boldsymbol{x}_j \sim p_d^-, \ j = 1, 2, \ldots, N\}$, and $\mathcal{D}^g = \{\boldsymbol{x}_k; \ \boldsymbol{x}_k \sim p_g, \ k = 1, 2, \ldots, N\}$, respectively, we train the Rumi-LSGAN models by replacing the expectations in the loss functions with their sample estimates as follows:

$$\mathcal{L}_D^{LS} = \frac{\beta^+}{N} \sum_{\boldsymbol{x}_i \in \mathcal{D}^+} \left( D(\boldsymbol{x}_i) - b^+ \right)^2 + \frac{\beta^-}{N} \sum_{\boldsymbol{x}_j \in \mathcal{D}^-} \left( D(\boldsymbol{x}_j) - b^- \right)^2 + \frac{1}{N} \sum_{\boldsymbol{x}_k \in \mathcal{D}^g} \left( D(\boldsymbol{x}_k) - a \right)^2, \text{ and}$$

$$\mathcal{L}_G^{LS} = \frac{\beta^+}{N} \sum_{\boldsymbol{x}_i \in \mathcal{D}^+} \left( D(\boldsymbol{x}_i) - c \right)^2 + \frac{\beta^-}{N} \sum_{\boldsymbol{x}_j \in \mathcal{D}^-} \left( D(\boldsymbol{x}_j) - c \right)^2 + \frac{1}{N} \sum_{\boldsymbol{x}_k \in \mathcal{D}^g} \left( D(\boldsymbol{x}_k) - c \right)^2.$$

We do not enforce the integral or non-negativity constraints explicitly, as it turns out that these are automatically satisfied during the optimization (cf. Supplementary material, Sections 1 and 2). A comparison between the Rumi-GAN variants is also provided in the supplementary document.

### 4.1   Evaluation Metrics

The Fréchet inception distance (FID) [36] is a useful metric for comparing the quality of the images generated by GANs. In our experiments, we use the InceptionV3 [37] model without the topmost layer, loaded with ImageNet pretrained weights to generate the embeddings over which the FID scores are evaluated. InceptionV3 requires a minimum input dimension of $76 \times 76 \times 3$. Hence, color images (CelebA, CIFAR-10) are rescaled to $80 \times 80 \times 3$ using bilinear interpolation. Gray-scale images (MNIST and Fashion MNIST) are rescaled to $80 \times 80$ and then replicated across three channels.

FID scores provide an objective assessment of the image quality, but not the diversity of the learnt distribution. Hence, we also report Precision-Recall (PR) behaviour, following the method of Sajjadi *et al.* [38]. The PR curve, in the context of generative models, gives the precision that a model of a given recall is likely to have, and vice versa. A high precision with a low recall implies that the model is able to generate samples of high quality in comparison with those coming from the target distribution, but at the cost of little diversity. On the other hand, a model that generates diverse images of low quality would have a low precision but a high recall. We use both FID scores and PR curves to analyze the performance of Rumi-LSGAN with respect to the baselines.

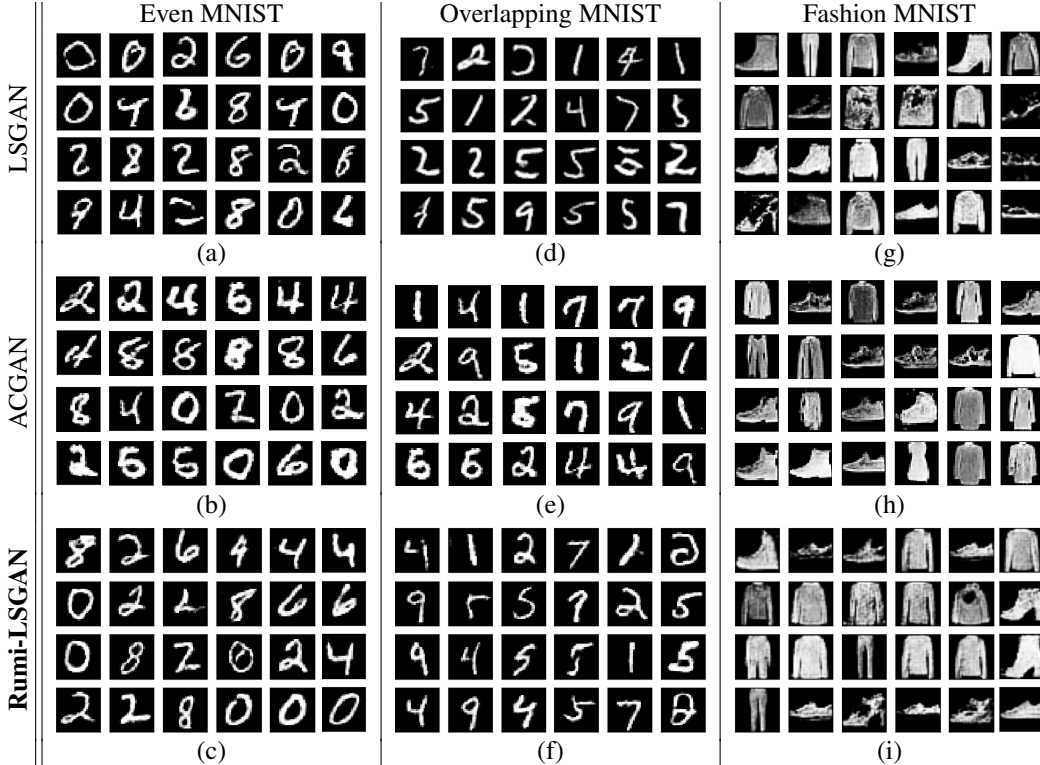

Figure 2: Comparison of the samples generated by LSGAN, ACGAN, and Rumi-LSGAN on: (a)-(c) MNIST with disjoint positive and negative classes; (d)-(f) MNIST with overlapping positive and negative classes; and (g)-(i) Fashion MNIST with overlapping positive and negative classes.

## 4.2 Experiments on MNIST and Fashion MNIST Datasets

***Experimental setup:*** Both baseline LSGAN and the Rumi-LSGAN use identical generator and discriminator networks. For ACGAN, the generator consists of an additional class label embedding on to a $7 \times 7 \times 1$ layer, which is concatenated with the noise input to the deconvolution layers. For comparison, the baseline model is trained only on the positive subset of data. The FID and PR performance measures are evaluated considering only positive samples as the reference dataset.

***Results:*** In the first experiment, we pool the five even digit classes of MNIST into the positive class, and the odd ones into the negative class. The LSGAN is trained solely on the positive class data, whereas the ACGAN and Rumi-LSGAN are trained using both positive and negative class data. Figures 2(a)-(c) show the images generated by the three models under comparison. We observe that the Rumi-LSGAN generates sharper images that consistently belong to the positive class unlike the other two models. This is evidence that the Rumi-LSGAN has a superior capability to learn to avoid the negative class. In the next experiment, we consider a configuration of the positive and negative classes with some overlap between them. More specifically, the positive ones are 1, 2, 4, 5, 7, and 9, and the negative ones are 0, 2, 3, 6, 8, and 9. The images generated in this configuration are shown in Figures 2(d)-(f). These images show that, despite the overlap between the positive and negative classes, Rumi-LSGAN consistently generates samples belonging to the positive class only, and of perceptually better quality. This experiment demonstrates that the positive class specification has an over-riding effect in the Rumi formulation, which is a desired feature.

Figure 3 compares the FID and PR curves of the three models. The shaded tolerance bands indicate the spread in the performance obtained over 10 runs. Rumi-LSGAN was found to settle at a lower FID in both the cases. ACGAN's relatively slower convergence of FID could be attributed to its more complex generation pipeline. We observe that the unmodified baseline LSGAN, trained only on the desired dataset achieves only average precision and recall values, with a minor improvement in the case of the six random classes, due to the additional samples available for training per epoch.

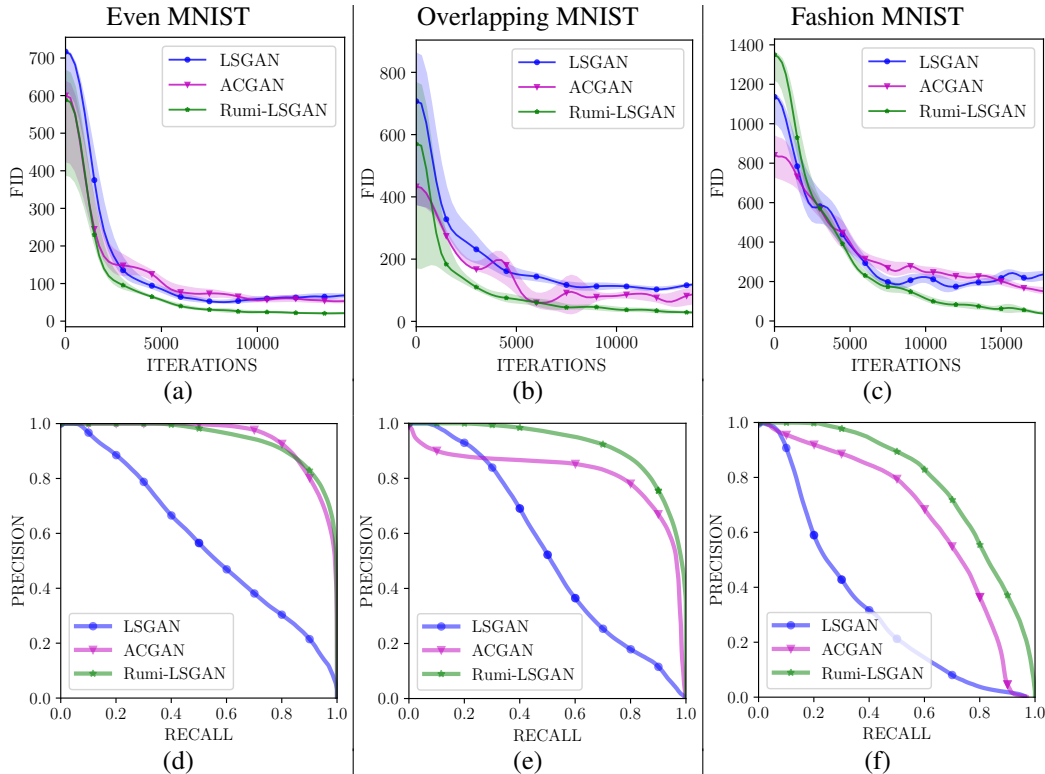

Figure 3: (🌀 Color Online) (a)-(c): A comparison of FID scores as iterations progress on: (a) Even numbers in MNIST; (b) Random subset of MNIST; and (c) Random subset of Fashion-MNIST. (d)-(f): Comparison of precision vs. recall on (d) Even numbers in MNIST; (e) Random subset of MNIST; and (f) Random subset of Fashion MNIST.

Rumi-LSGAN and ACGAN have comparable PR curves, with the Rumi-LSGAN achieving slightly better precision values due to its ability to strongly latch on to the desired positive class.

As an additional experiment, we show results on Fashion MNIST dataset with randomly picked positive and negative classes with overlap. Figures 2(g)-(i) show the images generated by the three GANs. Figures 3(c) and 3(f) show that Rumi-GAN converges relatively faster than the baselines with convergence measured in terms of FID and has better PR scores. The ACGAN has a poorer PR performance because it occasionally generates samples belonging to the wrong class. Additional comparisons on CelebA and CIFAR-10 datasets are presented in the supplementary document.

## 5 Handling Unbalanced Datasets

We now address a pertinent application of the Rumi-GAN framework — learning the minority class in unbalanced datasets. We simulate unbalanced data with MNIST, CelebA, and CIFAR-10 datasets by holding out samples from one of the classes and training the models to learn that class. In addition to the baselines considered in Section 4, we also compare Rumi-LSGAN with the twin auxiliary classifier GAN (TACGAN) [13] and CGANs with projection discriminator (CGAN-PD) [5], as they have been shown to generate balanced class distributions.

*Experimental Setup:* As an example, we consider digit 5 from the MNIST dataset to be the held-out class and the rest to be the negative classes. The imbalance is introduced by randomly picking 200 samples out of 6000 (less than $5\%$) exemplars of the digit 5 to constitute the minority positive class. The LSGAN is trained on only the 200 positive samples, whereas all other variants are trained on both positive and negative samples. Each model is trained for $10^4$ iterations. In the case of CelebA, we present results considering only $5\%$ of the images in the *Female* class to be the minority class, whereas from CIFAR-10, we demonstrate the performance when $5\%$ of the images in the *Horse* class are considered as the minority class. Since CelebA and CIFAR-10 datasets are more sophisticated

|                       || LSGAN | ACGAN | TAC-GAN | CGAN-PD | **Rumi-LSGAN** |
|-----------------------|--------|--------|----------|----------|----------------|
| MNIST (Averaged)      | 128.55 | 127.91 | 121.3    | 151.4    | **118.93**     |
| CelebA (Averaged)     | 226.25 | 243.95 | 213.6    | 281.51   | **169.34**     |
| CIFAR-10 (Averaged)   | 231.02 | 355.06 | 275.43   | 262.3    | **217.15**     |

Table 1: Comparison of FID scores on unbalanced datasets. Rumi-LSGAN has the best FID scores.

than MNIST, the models require more iterations to converge. Hence, we trained the models for $5 \times 10^4$ iterations on these datasets.

***Evaluation Metrics:*** We use FID scores and PR to compare the performance of Rumi-LSGAN with the baselines. Only the target class samples are queried from the conditional GAN variants. The FID scores and PR curves are compared with respect to the entire parent class of the held-out samples, which gives us a measure of how well the model has learnt to *generalize* and not simply *memorize* the training exemplars. We also tabulate the converged FID for all models in Table 1, averaged over multiple test cases. For MNIST and CIFAR-10, we average over results from choosing each class as the minority class. For CelebA, we average over the results obtained by choosing both *Males* and *Females* as instances of the under-represented class. We evaluate the FID upon convergence by drawing $10^4$ samples from the target class in the case of CelebA, and the entire target class in the case of MNIST and CIFAR-10 datasets. In all cases, $10^4$ samples are drawn from the generator.

***Results on MNIST Dataset:*** Figure 4 presents the samples learnt by the considered GAN variants. From Figures 4(a)-(e), we observe that the baseline LSGAN, which received no negative samples, performs better than ACGAN which collapsed on to a few classes (digits 3, 4, 6, and 8). Although TACGAN and CGAN-PD perform better than the baseline ACGAN, they also latch on to similar classes. Rumi-LSGAN produces visually more appealing samples than the baseline LSGAN. This is also reflected in the objective assessment carried out using the FID and PR scores shown in Figures 4(f) and 4(g). Rumi-LSGAN achieves a lower FID while also exhibiting higher precision and recall than the baseline LSGAN, thereby demonstrating that the model indeed generalized well. Conditional variants, on the other hand, perform poorly as they latch on to the majority classes.

***Results on CelebA and CIFAR-10:*** Similar results were obtained in the case of the CelebA and CIFAR-10 datasets. Mariani *et al.* [14] showed that the ACGAN suffers from mode collapse when trained on unbalanced data and our experiments also confirmed this behavior. Figures 4(h)-(l) and 4(o)-(s) show that the Rumi-LSGAN outperforms the baseline conditional models subjectively. In the case of CelebA, ACGAN, TACGAN, and CGAN-PD learn a mixture of both the desired and undesired classes, while on CIFAR-10, these models invariably latched on to the majority class. Table 1, Figures 4(m)-(n), and 4(t)-(u) further validate these observations through the convergence of FID scores and PR curves. While the converged models have relatively close FID scores, Rumi-LSGAN has the lowest FID score on both datasets. The PR curves indicate that Rumi-LSGAN generalized better to the desired held-out class. The poor PR performance of LSGAN may be attributed to mode collapse. Additional results are included in the supplementary document.

## 6   Conclusions

We introduced the Rumi formulation for GANs, which aims at generating samples from a desired positive class by learning to ignore the negative class, but with the training relying on samples coming from both the positive and the negative classes. We showed that the Rumi-SGAN generator can learn any weighted combination of the two classes, whereas the Rumi-LSGAN can specifically latch on to the positive class distribution. Validations on standard datasets such as MNIST, Fashion MNIST, CelebA and CIFAR-10 showed that Rumi-LSGAN outperforms the baseline and auxiliary-classifier based models particularly on learning a held-out class in unbalanced datasets. The validations serve to strengthen the philosophy that teaching the generator to avoid fitting to the undesired class distribution indeed boosts its performance on fitting to the desired class distribution. The Rumi framework is applicable to all known GAN flavors and regularized variants such as DRAGAN [39] and not just the standard GAN or LSGAN. For learning high-resolution images, models such as BigGAN [40], StyleGANs [41], or progressive growing of GANs (PGGANs) [42] can also be reformulated within the Rumi framework.

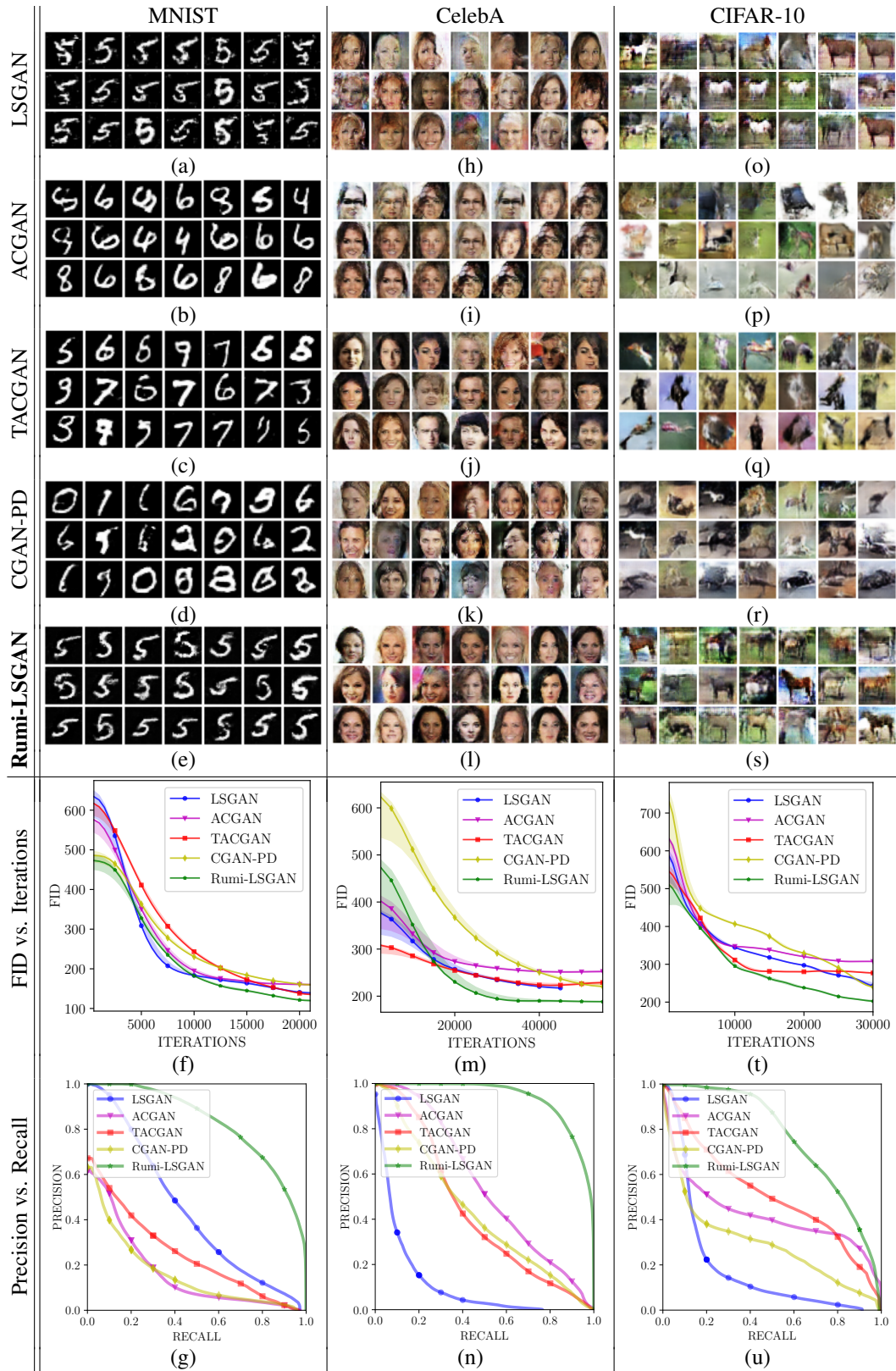

Figure 4: Results from training various GANs on unbalanced data: A comparison of generated samples on (a)-(e) MNIST with $5\%$ of digit class 5; (h)-(l) CelebA with $5\%$ of *Females* class; (o)-(s) CIFAR-10 with $5\%$ of *Horses* class. FID and PR curves from training the models on: (f) & (g) MNIST (digit class 5); (m) & (n) CelebA (Females); and (t) & (u) CIFAR-10 (*Horse* class). Rumi-LSGAN generates samples of superior quality, while learning only the distribution of the desired target class.

# 7 Acknowledgement

This work is supported by the Qualcomm Innovation Fellowship 2019.

# 8 Broader Impact

Neural network based image classification and supervised image generation are data-intensive tasks. These models, when trained on unbalanced data, for instance, facial image datasets with insufficient racial diversity [43], tend to inherit the implicit biases present in the data. DeVries *et al.* [44] demonstrated the existence of such biases with sub-par classification performance on images of objects coming from countries with low-income households, compared with those coming from countries with high-income households. The proposed approach could be used to address the imbalance in the data distribution and cater to the under-represented classes. The optimized generator in the proposed Rumi approach could be used to generate more samples of the under-represented classes and thus make the machine learning task more *inclusive*. The negative aspect is that one could flip the whole argument around and redefine the desired and undesired classes to serve exactly the opposite objective of favoring a certain class at the expense of the other. While the proposed approach could be used to alleviate the imbalances and biases present in the dataset, it cannot overcome the biases of the data scientist.

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
