[Supplementary Material]

# Teaching a GAN What Not to Learn
# (Supplementary Material)

**Siddarth Asokan**[*]
Robert Bosch Center for Cyber-Physical Systems
Indian Institute of Science
Bangalore, India
siddartha@iisc.ac.in

**Chandra Sekhar Seelamantula**
Department of Electrical Engineering
Indian Institute of Science
Bangalore, India
css@iisc.ac.in

## Overview

We provide additional analytical and experimental results to support the content presented in the main manuscript. Section 1 of this document presents a detailed discussion on the Rumi-LSGAN. In Section 2, we impose the Rumi formulation on $f$-GANs [1], and in Section 3, we generalize it to include integral probability metric (IPM) based GANs such as the Wasserstein GAN (WGAN) [2]. In Section 4, we compare the performance of Rumi-SGAN, Rumi-LSGAN, and Rumi-WGAN on the MNIST dataset. Finally, in Section 5, we provide additional results and comparisons on CelebA and CIFAR-10 datasets.

## 1 Rumi-LSGAN

Recall the Rumi-LSGAN formulation:

$$\mathcal{L}_D^{LS} = \beta^+ \, \mathbb{E}_{\boldsymbol{x} \sim p_d^+} \left[ \left( D(\boldsymbol{x}) - b^+ \right)^2 \right] + \beta^- \, \mathbb{E}_{\boldsymbol{x} \sim p_d^-} \left[ \left( D(\boldsymbol{x}) - b^- \right)^2 \right] + \mathbb{E}_{\boldsymbol{x} \sim p_g} \left[ \left( D(\boldsymbol{x}) - a \right)^2 \right],$$

$$\mathcal{L}_G^{LS} = \beta^+ \, \mathbb{E}_{\boldsymbol{x} \sim p_d^+} \left[ \left( D^*(\boldsymbol{x}) - c \right)^2 \right] + \beta^- \, \mathbb{E}_{\boldsymbol{x} \sim p_d^-} \left[ \left( D^*(\boldsymbol{x}) - c \right)^2 \right] + \mathbb{E}_{\boldsymbol{x} \sim p_g} \left[ \left( D^*(\boldsymbol{x}) - c \right)^2 \right],$$

where $\mathcal{L}_G^{LS}$ must be subjected to the integral and non-negativity constraints

$$\Omega_{p_g} : \int_{\mathcal{X} \subseteq \mathbb{R}^n} p_g(\boldsymbol{x}) \, \mathrm{d}\boldsymbol{x} = 1, \qquad \text{and} \qquad \Phi_{p_g} : p_g(\boldsymbol{x}) \geq 0, \, \forall \, \boldsymbol{x},$$

respectively. Incorporating the constraints using a Lagrangian formulation and expressing the expectations as integrals results in

$$\mathcal{L}_D^{LS} = \int_{\mathcal{X}} \left( \beta^+ \left( D(\boldsymbol{x}) - b^+ \right)^2 p_d^+ + \beta^- \left( D(\boldsymbol{x}) - b^- \right)^2 p_d^- + \left( D(\boldsymbol{x}) - a \right)^2 p_g \right) \mathrm{d}\boldsymbol{x}, \text{ and} \quad (1)$$

$$\mathcal{L}_G^{LS} = \int_{\mathcal{X}} \left( \left( D^*(\boldsymbol{x}) - c \right)^2 \left( \beta^+ p_d^+ + \beta^- p_d^- + p_g \right) + \lambda_p p_g + \mu_p(\boldsymbol{x}) p_g \right) \mathrm{d}\boldsymbol{x} - \lambda_p \,, \quad (2)$$

where $\lambda_p$ and $\mu_p(\boldsymbol{x})$ are the Karush-Kuhn-Tucker (KKT) multipliers. The cost functions in Equations (1) and (2) have to be optimized with respect to the discriminator $D$ and the generator $p_g$, respectively. Since it is a functional optimization problem, we have to invoke the *Calculus of Variations*. If the integrand is continuously differentiable everywhere over the support $\mathcal{X} = \text{Supp}(p_d^+) \cup \text{Supp}(p_d^-) \cup \text{Supp}(p_g) \subseteq \mathbb{R}^n$, then the optimization of the integral cost carries over point-wise to the integrand. The optimal discriminator turns out to be

$$D^*(\boldsymbol{x}) = \frac{b^+ \beta^+ p_d^+ + b^- \beta^- p_d^- + a p_g}{\beta^+ p_d^+ + \beta^- p_d^- + p_g}.$$

---

[*]Corresponding Author

The optimal generator turns out to be the solution to a quadratic equation with roots

$$p_g^* = \beta^+ \left( \frac{\pm(a - b^+)}{\sqrt{(a - c)^2 + \lambda_p + \mu_p}} - 1 \right) p_d^+ + \beta^- \left( \frac{\pm(a - b^-)}{\sqrt{(a - c)^2 + \lambda_p + \mu_p}} - 1 \right) p_d^-. \quad (3)$$

The positive root is the minimizer of the cost. The optimal KKT multiplier $\mu_p^*(x)$ is the one that satisfies the complementary slackness condition $\mu_p^* p_g^* = 0$, $\forall\, x \in \mathcal{X}$, and the feasibility criterion $\mu_p^*(x) \leq 0$, $\forall x \in \mathcal{X}$. Since $p_g^*$ is a weighted mixture of $p_d^+$ and $p_d^-$, the support of the solution could be split into three regions corresponding to: (i) $p_d^+ > 0$ and $p_d^- > 0$; (ii) $p_d^+ > 0$ and $p_d^- = 0$; and (iii) $p_d^+ = 0$ and $p_d^- > 0$. Enforcing the complementary slackness condition in each region, with suitable assumptions on the class labels and weights such as $a \leq \frac{b^+ + b^-}{2}$ and $\beta^+ > 0, \beta^- > 0$, yields $\mu_p^*(x) = 0$, $\forall x \in \mathcal{X}$, as the only feasible solution with a consistent value for $\lambda_p^*$ over all three regions constituting the support. Enforcing the integral constraint $\Omega_{p_g}$ and solving for $\lambda_p^*$ gives

$$\lambda_p^* = \left( \frac{\beta^+(a - b^+) + \beta^-(a - b^-)}{1 + \beta^+ + \beta^-} \right)^2 - (a - c)^2.$$

Substituting for $\mu_p^*$ and $\lambda_p^*$ in (3) yields the optimal generator:

$$p_g^*(x) = \beta^+ \eta^+ p_d^+(x) + \beta^- \eta^- p_d^-(x),$$

where

$$\eta^+ = \left( \frac{(1 + \beta^-)(a - b^+) - \beta^-(a - b^-)}{\beta^+(a - b^+) + \beta^-(a - b^-)} \right),$$

and

$$\eta^- = \left( \frac{(1 + \beta^+)(a - b^-) - \beta^+(a - b^+)}{\beta^+(a - b^+) + \beta^-(a - b^-)} \right).$$

The solutions for $\mu_p^*$ and $\lambda_p^*$ are also intuitively satisfying as the optimal generator obtained under these conditions automatically satisfies the non-negativity constraint. With this choice of the class labels and weights, the solution obtained by applying only the integral constraint $\Omega_{p_g}$ automatically satisfies the non-negativity constraint.

As a special case, observe that setting $\beta^+ = \frac{a - b^-}{b^- - b^+}$ results in $\eta^- = 0$. Subsequently, any choice of $a, b^+, b^-$, and $\beta^-$ such that $\beta^+ \eta^+ = 1$ gives $p_g^* = p_d^+$. Similarly, setting $\beta^- = \frac{a - b^+}{b^+ - b^-}$ and $\beta^- \eta^- = 1$ yields $p_g^* = p_d^-$. Lastly, when $a = \frac{b^+ + b^-}{2}$, we have

$$p_g^* = \left( \frac{\beta^+(1 - 2\beta^-)}{\beta^+ + \beta^-} \right) p_d^+ + \left( \frac{\beta^-(1 + 2\beta^+)}{\beta^+ + \beta^-} \right) p_d^-,$$

which is a mixture of $p_d^+$ and $p_d^-$.

## 2   The Rumi-$f$-GANs

The Rumi formulation is extendable to all the $f$-GAN variants presented by Nowozin *et al.* [1]. Consider a GAN that minimizes the generalized divergence metric

$$D_f(p_g, p_d) = \int_{\mathcal{X}} p_d(x) f \left( \frac{p_g(x)}{p_d(x)} \right) dx,$$

which was shown to be equivalent to minimizing the loss functions [1]:

$$\mathcal{L}_D^f = -\mathbb{E}_{x \sim p_d}[T(x)] + \mathbb{E}_{x \sim p_g}[f^c(T(x))], \quad \text{and}$$

$$\mathcal{L}_G^f = \mathbb{E}_{x \sim p_d}[T(x)] - \mathbb{E}_{x \sim p_g}[f^c(T(x))],$$

with respect to $D(x)$ and $p_g$, respectively, where $f^c$ is the Fenchel conjugate of the divergence $f$, and $T(x) = g(D(x))$ with $g$ explicitly representing the activation function employed at the output

of the discriminator network. The Rumi-$f$-GAN minimizes $D_f^+ = D_f(p_g, p_d^+)$, while maximizing $D_f^- = D_f(p_g, p_d^-)$, weighted by $\gamma^+$ and $\gamma^-$, respectively, which is given as

$$\mathcal{L}_D^{Rf} = -\gamma^+ \mathbb{E}_{\boldsymbol{x} \sim p_d^+}[T(\boldsymbol{x})] + \gamma^- \mathbb{E}_{\boldsymbol{x} \sim p_d^-}[T(\boldsymbol{x})] + \mathbb{E}_{\boldsymbol{x} \sim p_g}[f^c(T(\boldsymbol{x}))], \text{ and} \tag{4}$$

$$\mathcal{L}_G^{Rf} = \gamma^+ \mathbb{E}_{\boldsymbol{x} \sim p_d^+}[T(\boldsymbol{x})] - \gamma^- \mathbb{E}_{\boldsymbol{x} \sim p_d^-}[T(\boldsymbol{x})] - \mathbb{E}_{\boldsymbol{x} \sim p_g}[f^c(T(\boldsymbol{x}))], \tag{5}$$

where $\gamma^+ - \gamma^- = 1$, and $\mathcal{L}_G^{Rf}$ is subjected to the integral and non-negativity constraints $\Omega_{p_g}$ and $\Phi_{p_g}$, respectively. As shown in Section 1, we enforce only $\Omega_{p_g}$, showing that the optimal solution satisfies $\Phi_{p_g}$ without having to enforce it explicitly.

**Lemma 2.1.** *The optimal Rumi-$f$-GAN: Consider the $f$-GAN optimization problem defined through Equations (4) and (5). Assume that the weights satisfy $\gamma^+ - \gamma^- = 1$. The optimal discriminator and generator are the solutions to*

$$\frac{\partial f^c}{\partial T} = \frac{\gamma^+ p_d^+ - \gamma^- p_d^-}{p_g}, \text{ and} \tag{6}$$

$$f^c(T^*) = \lambda_p, \tag{7}$$

*respectively, where $T^* = g(D^*)$.*

**Proof:** The integral Rumi-$f$-GAN costs can be optimized as in the case of Rumi-LSGAN. Optimization of the integrand in Equation (4) yields the necessary condition that the optimal discriminator $D^*(\boldsymbol{x})$ must satisfy, which gives us Equation (6). Differentiating the integrand in (5), we get

$$\left( (\gamma^+ p_d^+ - \gamma^- p_d^-) - p_g \frac{\partial f^c}{\partial T^*} \right) \frac{\partial T^*}{\partial p_g} - f^c(T^*) + \lambda_p = 0.$$

Enforcing the condition given in (6) yields the necessary condition that $p_g^*$ must satisfy. $\qquad \square$

Table 1 shows the optimal discriminator and generator functions obtained in the case of each of the $f$-GAN variants presented by Nowozin *et al.* [1]. We observe that all $f$-GANs learn a weighted mixture of $p_d^+$ and $p_d^-$, akin to the Rumi-SGAN presented in the main manuscript. The weights are a function of $\gamma^+, \gamma^-$, and $\lambda_p$. The optimal $\lambda_p^*$ in each case can be found by enforcing the integral constraint $\Omega_{p_g}$. Also observe that the Rumi-Pearson-$\chi^2$ GAN in Table 1 is a special case of the Rumi-LSGAN, where $a - c = 1$, $(a - b^+) = \sqrt{\lambda_p + 1} + 1$ and $(a - b^-) = \sqrt{\lambda_p + 1} - 1$.

Finally, we note that setting $\gamma^+ \in [0, 1]$ in addition to $\gamma^+ - \gamma^- = 1$ result in solutions for all $f$-GANs that automatically satisfy the non-negativity constraint.

Table 1: Rumi formulation of $f$-GANs: The optimal Rumi discriminator $D^*$ and generator $p_g^*$ for a given $f$-GAN defined through its activation function $g$ and Fenchel conjugate $f^c$ of the divergence metric. $T = g(D)$.

| $f$-divergence | $g(D)$ | $f^c(T)$ | $D^*(\boldsymbol{x})$ | $p_g^*(\boldsymbol{x})$ |
|---|---|---|---|---|
| Kullback-Leibler (KL) | $D$ | $e^{T-1}$ | $1 + \log\left(\frac{\gamma^+ p_d^+ - \gamma^- p_d^-}{p_g}\right)$ | $\frac{\gamma^+}{\log(\lambda_p)} p_d^+ - \frac{\gamma^-}{\log(\lambda_p)} p_d^-$ |
| Reverse KL | $-e^{-D}$ | $-1 - \log(-T)$ | $\log\left(\frac{\gamma^+ p_d^+ - \gamma^- p_d^-}{p_g}\right)$ | $\frac{\gamma^+}{e^{\lambda_p+1}} p_d^+ - \frac{\gamma^-}{e^{\lambda_p+1}} p_d^-$ |
| Pearson-$\chi^2$ | $D$ | $\frac{1}{4}T^2 + T$ | $2\left(\frac{\gamma^+ p_d^+ - \gamma^- p_d^- - p_g}{p_g}\right)$ | $\frac{\gamma^+}{\sqrt{\lambda_p+1}} p_d^+ - \frac{\gamma^-}{\sqrt{\lambda_p+1}} p_d^-$ |
| Squared-Hellinger | $1 - e^{-D}$ | $\frac{T}{1-T}$ | $\frac{1}{2} \log\left(\frac{\gamma^+ p_d^+ - \gamma^- p_d^-}{p_g}\right)$ | $\frac{\gamma^+}{(\lambda_p+1)^2} p_d^+ - \frac{\gamma^-}{(\lambda_p+1)^2} p_d^-$ |
| SGAN | $-\log\left(1 - e^{-D}\right)$ | $-\log\left(1 - e^T\right)$ | $\log\left(\frac{\gamma^- p_d^- - \gamma^+ p_d^+}{p_g}\right)$ | $\frac{\gamma^+ \lambda_p}{1-\lambda_p} p_d^+ - \frac{\gamma^- \lambda_p}{1-\lambda_p} p_d^-$ |

## 3 Rumi-WGAN

All integral probability metric (IPM) based GANs [2–4] can be reformulated under the Rumi framework. As an example, we consider the Rumi flavor of the Wasserstein GAN (Rumi-WGAN). The Rumi-WGAN minimizes the earth-mover distance (EMD) between $p_d^+$ and $p_g$, while maximizing the EMD between $p_d^-$ and $p_g$. Both these terms can be independently brought to the Kantarovich-Rubinstein dual-form as in WGANs [2]:

$$D^* = \underset{D, \|D\|_L \leq 1}{\arg\max} \left( \gamma^+ \left( \mathbb{E}_{\boldsymbol{x} \sim p_d^+}[D(\boldsymbol{x})] - \mathbb{E}_{\boldsymbol{x} \sim p_g}[D(\boldsymbol{x})] \right) + \gamma^- \left( \mathbb{E}_{\boldsymbol{x} \sim p_d^-}[D(\boldsymbol{x})] - \mathbb{E}_{\boldsymbol{x} \sim p_g}[D(\boldsymbol{x})] \right) \right),$$

$$p_g^* = \underset{p_g}{\arg\min} \left( \gamma^+ \left( \mathbb{E}_{\boldsymbol{x} \sim p_d^+}[D(\boldsymbol{x})] - \mathbb{E}_{\boldsymbol{x} \sim p_g}[D(\boldsymbol{x})] \right) - \gamma^- \left( \mathbb{E}_{\boldsymbol{x} \sim p_d^-}[D(\boldsymbol{x})] - \mathbb{E}_{\boldsymbol{x} \sim p_g}[D(\boldsymbol{x})] \right) \right),$$

which directly result in the Rumi-WGAN costs:

$$\mathcal{L}_D^W = -\gamma^+ \left( \mathbb{E}_{\boldsymbol{x} \sim p_d^+}[D(\boldsymbol{x})] - \mathbb{E}_{\boldsymbol{x} \sim p_g}[D(\boldsymbol{x})] \right) - \gamma^- \left( \mathbb{E}_{\boldsymbol{x} \sim p_d^-}[D(\boldsymbol{x})] - \mathbb{E}_{\boldsymbol{x} \sim p_g}[D(\boldsymbol{x})] \right), \text{ and}$$

$$\mathcal{L}_G^W = \gamma^+ \left( \mathbb{E}_{\boldsymbol{x} \sim p_d^+}[D^*(\boldsymbol{x})] - \mathbb{E}_{\boldsymbol{x} \sim p_g}[D^*(\boldsymbol{x})] \right) - \gamma^- \left( \mathbb{E}_{\boldsymbol{x} \sim p_d^-}[D^*(\boldsymbol{x})] - \mathbb{E}_{\boldsymbol{x} \sim p_g}[D^*(\boldsymbol{x})] \right),$$

with the constraint that $D(\boldsymbol{x})$ is Lipschitz-1. Similar to WGAN-GP [5], we enforce the Lipschitz constraint through the gradient penalty: $(\|\nabla_{\boldsymbol{x}} D(\boldsymbol{x})\|_2 - 1)^2$, which is evaluated in practice by a sum over points interpolated between $p_d^+$ and $p_g$, and $p_d^-$ and $p_g$:

$$\Omega_{GP} : \sum_{\hat{\boldsymbol{x}}} (\|\nabla_{\hat{\boldsymbol{x}}} D(\hat{\boldsymbol{x}})\|_2 - 1)^2 + \sum_{\tilde{\boldsymbol{x}}} (\|\nabla_{\tilde{\boldsymbol{x}}} D(\tilde{\boldsymbol{x}})\|_2 - 1)^2,$$

where $\hat{\boldsymbol{x}} = (1 - \xi)\boldsymbol{x}^g + \xi\boldsymbol{x}^+$ and $\tilde{\boldsymbol{x}} = (1 - \zeta)\boldsymbol{x}^g + \zeta\boldsymbol{x}^-$ such that $\boldsymbol{x}^g \sim p_g, \boldsymbol{x}^+ \sim p_d^+, \boldsymbol{x}^- \sim p_d^-$, and $\xi, \zeta$ are uniformly distributed over $[0, 1]$.

## 4 Comparison of Rumi-GAN Variants

In this section, we compare the performance of Rumi-SGAN, Rumi-LSGAN, and Rumi-WGAN-GP with their baseline variants on the MNIST dataset. We consider the following test scenarios:

1. Even digits as the positive class;
2. Overlapping positive and negative classes as described in the main manuscript (Section 4.2);
3. One vs. rest learning where the positive class comprises all samples from the digit class 5, with the rest of the digit classes representing the negative class data.

Scenario 3 above simulates an important variant of learning from unbalanced data.

***Experimental Setup:*** The network architectures and hyper-parameters are as described in the main manuscript (Section 4). For Rumi-SGANs, we set $\alpha^+ = 0.8$ and $\alpha^- = -0.2$. Rumi-LSGAN uses class labels $(a, b^-, c, b^+) = (0, 0.5, 1, 2)$ with weights $\beta^+ = 1$ and $\beta^- = 0.5$. For Rumi-WGAN-GP, we use $\gamma^+ = 5$ and $\gamma^- = 1$.

***Results:*** Figures 1, 2, and 3 show the samples generated by the various GANs under consideration. From Figure 1, we observe that the Rumi formulation always results in an improvement in the visual quality of the images generated. In the case of SGAN, the baseline approach experienced mode collapse (Fig. 1(a)), while its Rumi counterpart (Fig. 1(b)) learnt the target distribution accurately. Observe that the Rumi-SGAN and Rumi-WGAN variants learn to generate a few random samples from the negative class as well — this validates our claim that these variants always learn a mixture of the positive and negative class densities. Similar improvements in the visual quality of images are also seen in Scenario 2, which considers overlapping classes (Fig. 2), or Scenario 3, which considers *one vs. the rest* learning (Fig. 3).

The Fréchet inception distance (FID) plots and precision-recall (PR) curves in Figure 4 quantitatively validate our findings. The Rumi variants achieve a higher precision and recall, and saturate to better FID values than their respective baselines. Also, Rumi-SGAN and Rumi-LSGAN show equivalent performance. As training the LSGAN is relatively more stable than training the SGAN, we prefer the Rumi-LSGAN to Rumi-SGAN when considering complex datasets such as CelebA.

**Baseline**                    **Rumi counterpart**

(a) SGAN

(b) Rumi-SGAN

(c) LSGAN

(d) Rumi-LSGAN

(e) WGAN

(f) Rumi-WGAN

Figure 1: (⬤ Color Online) Illustrating the strength of the *Rumi framework*. **Scenario 1**: Training GANs to generate even digits from the MNIST dataset. The Rumi counterparts learn qualitatively better images than the corresponding baselines. The SGAN seems to have mode-collapsed unlike its Rumi counterpart.

**Baseline**                    **Rumi counterpart**

(a) SGAN

(b) Rumi-SGAN

(c) LSGAN

(d) Rumi-LSGAN

(e) WGAN

(f) Rumi-WGAN

Figure 2: (⬤ Color Online) **Scenario 2**: Training GANs on overlapping MNIST classes. The Rumi counterparts generate visually better images than the baselines. The *mode-collapse* effect is prominent in the baselines unlike the Rumi counterparts.

Figure 3: (🎨 Color Online) **Scenario 3**: Training GANs to learn the digit class 5. The Rumi variants generate sharper images compared with the corresponding baselines.

Figure 4: (🎨 Color Online) Comparison of FID vs. iterations and PR curves for various GANs trained on the MNIST dataset with positive class data being: (a) & (b) Even numbers; (c) & (d) Overlapping subsets; and (e) & (f) Single digit class 5. Rumi variants possess better precision and recall characteristics, and also achieve better FID values than the baseline models.

# 5 Validation on CIFAR-10 and CelebA Datasets

We now present additional experimental results on CIFAR-10 and CelebA datasets. The experimental setup is identical to that explained in the main manuscript (Sections 4 and 5). All the models are trained for $10^5$ iterations. CelebA images are rescaled to $32 \times 32 \times 3$ unless stated otherwise.

***Experimental Setup:*** On the CIFAR-10 dataset, we consider the case of learning disjoint positive and negative classes. The *animal* classes are labelled as positive, and the *vehicle* classes as negative. This is a scenario where the negative class samples have very little resemblance to those in the positive class. In the main manuscript, for CelebA, we presented results of learning the class of *female* celebrities as the positive class while setting the *males* to be the negative class. Here, we consider the converse situation — positive class of *male celebrities* and the negative class of *female celebrities*. We also present additional experimental results on learning minority classes in CelebA. Splitting the data based on the *bald* or the *hat* class label of CelebA gives about 5,000 positive samples and 195,000 negative class samples in each case.

***Results:*** Figure 5 presents the samples generated by SGAN, LSGAN, and their Rumi counterparts, alongside those generated by sampling an ACGAN trained purely on the *animal classes* of CIFAR-10. We observe that the Rumi-SGAN and Rumi-LSGAN generate images of superior visual quality. The Rumi variants also have better FID and PR performance than the baselines (Figures 8(a) & (b)). From Figure 5(b), we observe that, in the context of Rumi-SGAN, no visually discernible features from the negative class of *vehicles* are present. This shows that although, in principle, the optimal Rumi-SGAN learns a mixture of the positive and negative classes, in practice, on real-world image datasets such as CIFAR-10, the influence of the negative class in the mixture is negligible.

The images generated by LSGAN, Rumi-LSGAN and ACGAN on CelebA are given in Figure 6. Rumi-LSGAN generates visually better images than the baseline LSGAN and ACGAN in all the three cases considered. In the case on unbalanced data, the ACGAN latches on to the more-represented negative class. whereas Rumi-LSGAN is able to generate images exclusively from the target positive class. The FID and PR curves are presented in Figures 8(c)-(f). In the case of unbalanced data, although ACGAN has a comparable PR score as that of Rumi-LSGAN, a majority of the samples generated correspond to the undesired (negative) class — the positive/negative class label is something that the embeddings used to evaluate the PR measure are oblivious to. We attribute the relatively poorer PR performance of all models on the CelebA *Bald* dataset to insufficient reference positive class samples, resulting in poorer estimates of the model statistics.

Finally, we show results on high-resolution CelebA images ($128 \times 128 \times 3$). We train Rumi-LSGAN on both the simulated unbalanced class data (*Females*/*Males* classes with $5\%$ positive samples and the other class negative) and true unbalanced classes (*Bald* and *Hat* classes). From the results shown in Figure 7, we observe that Rumi-LSGAN generalizes well to the high-resolution scenario as well, generating a diverse set of images from the desired positive class in all cases.

## Source Code

The TensorFlow source code and models for all the experiments reported in this paper are available at the following GitHub Repository: `https://github.com/DarthSid95/RumiGANs.git`.

Figure 5: (● Color Online) A comparison of the samples generated by the various GANs on *animals* from CIFAR-10. The Rumi counterparts generate qualitatively better images than the baselines.

Figure 6: ( Color Online) A comparison of the samples generated by various GANs on the CelebA dataset with positive samples drawn from: (a)-(c) the class of *male* celebrities; (d)-(f) the class of *bald* celebrities; (g)-(i) the class of celebrities wearing a *hat*. On unbalanced data, ACGAN latched onto the majority classes, while Rumi-LSGAN generated samples from the correct class.

Figure 7: (🌀 Color Online) High-resolution CelebA images generated by Rumi-LSGAN.

Figure 8: (🌀 Color Online) Comparison of FID vs. iterations and PR curves of various GANs when the training is carried out on: (a) & (b) CIFAR-10 *animal* classes; (c) & (d) CelebA *male* class; and (e) & (f) CelebA *bald* class. Rumi-LSGAN outperforms the baselines in terms of FID and PR values.