[Reviews · NeurIPS 2020]

Review 1

Summary and Contributions: This paper proposed to improve GAN by incorporating negative samples. The proposed Rumi-GAN can be easily applied to another GAN-based framework, which only requires to divide training data into the positive and negative ones. The author further compared with ACGAN, LSGAN, and the modified Rumi-LSGAN on several most popular dataset.

Strengths: The method is simple yet framework, it gives good results compared with the baseline model. Sufficient experiments are constructed to show the efficiency of the proposed method, which helps stabilize the GAN training and a better FID performance. The theoretical analysis is straightforward and clear.

Weaknesses: 1)The intuition behind it is not well conveyed. Why do we need to build a third negative class? The problem is not well defined in this paper. 2) It would be unfair to compare with ACGAN. ACGAN has an inherent issue which causes a bad performance (referred in many papers such as TAC-GAN[1]). 3) The iterations plot in Figure 3 for GAN training actually does not help explain the model efficiency. The final Fid score, converging speed, and sensitivity of the hyper-parameters (alpha) would be interesting. 4) there might be some typo errors in the caption of Figure 1. (Color Online ). 5) The contribution of this paper is explained, but the novelty is not sufficient. The problem setup of this paper seems is not mentioned. [1]Gong, M., Xu, Y., Li, C., Zhang, K., and Batmanghelich, K. Twin auxilary classifiers gan. Advances in Neural Information Processing Systems 32, pp.1328–1337. Curran Associates, Inc., 2019b

Correctness: The formulation of the proposed method is correct with the theoretical analysis. The empirical methodology is correct.

Clarity: Motivation of the paper is not clear. There are some minor typo errors; the rest of the paper looks good.

Relation to Prior Work: what is the relationship between this wok and many works in complementary learning (there are GAN version of those as well).

Reproducibility: Yes

Additional Feedback:


Review 2

Summary and Contributions: This paper propose to train GANs by splitting the data distribution into two subset: positive parts and negative parts. By adjusting the objective functions to Eqn. (1)&(2) or (3)&(4), they get the Rumi GAN’s formulation for SGAN or LSGAN.

Strengths: The idea is somehow novel.

Weaknesses: The presentation of this paper is not clear. First, I’m not sure about the main target of this paper. Is it targeting on improve the performance of GANs, or just on getting a useful discriminator. Besides, the motivation is not clearly presented. Second, the setting of the problem is not clear. The data distribution is split into p_{d}^+ and p_{d}^-. Then why to split it? It is split according to what? How to split it in real world dataset such as MNIST? Third, what’s the meaning of the regularization term in L94? Why the regularization disappears for LSGAN? Since the generator is a mapping from a random variable z to the image space, the distributions of G defined by this mapping should automatically satisfy these regularizations.

Correctness: The presentation is ambiguous and I can hardly verify the correctness of this paper.

Clarity: The presentation can be improved. See details in the weakness.

Relation to Prior Work: Yes, at least to my knowledge.

Reproducibility: Yes

Additional Feedback: Maybe I made a misunderstanding about this paper. I'm looking forward to the author's feedback to improve the clarity of this paper. Post Rebuttal: After the rebuttal, I carefully read the paper again and discussed it with other reviewers. I acknowledge the novelty of this paper. Since this paper consider a special case of incorporating the label information to boost the performance of conditional generation, comparing the proposed method with the SOTA cGANs is necessary, such as the discriminator with conditional projection [1*]. Comparing it with ACGAN is somehow weak. I vote for conditional acceptance if the authors can make a systematic comparison to the SOTA results. Besides, I'm not sure how the FID score is evaluated. For the unconditional setting, the FID can achieve under 30 on the CelebA dataset. I guess that maybe the authors compare the statistics of the generated samples with the statistics of the whole dataset rather than the statistics of positive samples. The Intra-FID proposed in [1*] is a better evaluation metric for this setting. [1*] Miyato, Takeru, and Masanori Koyama. "cGANs with projection discriminator." arXiv preprint arXiv:1802.05637 (2018).


Review 3

Summary and Contributions: This paper proposes a way to leverage negative samples during GAN training. They show how their method can be plugged in existing GAN frameworks, and demonstrate relative improvement over such frameworks on MNIST (28x28), CIFAR (32x32) and rescaled CelebA (32x32) in terms of FID and Precision/Recall curves, and in the case where the positive classes represent subsets of the original datasets (negative samples being the remaining).

Strengths: - The proposed solution is light, and can be added to existing frameworks with little additional computational complexity (only adds some terms to existing losses). The method introduces hyperparameters but they are fixed for all datasets, and insights from the theoretical analysis already bounds their possible values. - For both instanciations of the method (SGAN and LSGAN), the optimal discriminator and generator are explicitly derived and analyzed, which gives a clear understanding of how negative sampling affects the obtained generated distribution. - Experimental results look promising in the evaluated setting, showing consistent improvement over LSGAN and ACGAN in terms of FID and P/R curve. - Paper is fairly well written and easy to follow

Weaknesses: - The idea of using negative samples is not new and is still widely used in metric learning (triplet loss, contrastive loss etc..) / representation learning (Discriminative unsupervised feature learning, Representation learning with contrastive predictive coding etc..). Even though the setting is different here, I think the related work section should at least mention and discuss some of those works. - I randomly checked 1 mathematical derivation (l.130), and it was incorrect. The beta_+ given does not allow to set eta_- to 0 (in fact the solution would rather be 1/beta_+). Hence I wonder how this value was set during experiments (not discussed after). - I found some experiments confusing. For instance, the even and overlapping MNIST. Why should ACGAN P/R curves be any different between the two experiments: ACGAN is trained on the full MNIST dataset regardless of the positive/negative split, isn'it ? Even if the set of digits over which you evaluate the scores may slightly differ in both cases, this shouldn't make such a difference on MNIST. - Experiments are exclusively carried out on low-resolution images (at most 32x32), which makes it hard to evaluate how the method would perform in more "realistic" settings. - I also wonder how the method would behave if a domain shift was also introduced between positive and negative classes : say you include CIFAR images as negative samples when learning classes from CelebA. Would the performances be enhanced, or could they be degraded w.r.t a vanilla LSGAN trained on positive samples only ?

Correctness: I didn't check all mathematical derivations in detail, but the results of lemmas appear intuitively correct. As for the experiments, the methodology appears overall correct. However, there is no detail on why certain particular subsets are chosen for CIFAR and CelebA (e.g horses and female). Given the results in supplementary, it seems results can vary a lot according to the positive group chosen (c.f Figure 7 between "Males" and "Bald" groups). Hence I think presenting results averaged over several randomly chosen classes would be more stastically significant.

Clarity: Yes the paper is well written.

Relation to Prior Work: See the above comments related to the weaknesses.

Reproducibility: No

Additional Feedback:


Review 4

Summary and Contributions: The paper presents a new GAN framework, Rumi-GAN for positive data generation while having access to both positive data and negative data.

Strengths: Quality: The paper the technically sound. The Lemma’s are well explained. This work is a complete piece. In section 2, the author claims any flavor of GAN could be accommodated within this framework. I am very glad that in the supplementary the author provides the accommodation for f-GAN and integral probability metric based GAN like Wasserstein GAN. Great job! The experiment part also has good quality. Comparison with SOTA baselines, experiments on different datasets, and different settings in one dataset are delivered. Originality: The main contribution of this paper is to introduce the principle “the art of knowing is knowing what to ignore” to data generation. It is quite interesting that one can leverage negative data samples to help positive data generation. The most related work is GenPU[0] which also introduces the concept of positive/negative data generation. But GenPU is more focused on discriminative learning. While Rumi-GAN is focused on generative learning. So to my knowledge, this paper has good originality. Significance: Learning a generative model for imbalanced data or for data with very few examples are important machine learning tasks. This work provides a promising solution by leveraging other untargeted and similar data (called negative data here). The methodology sounds practical to me. Hope it can benefit real applications. [0] Generative Adversarial Positive-Unlabelled Learning

Weaknesses: In general, it is a good paper to me. No major issues.

Correctness: I did not check every proof. But the lemma seems reasonable to me. The empirical study is quite good. The goal of every subtask clear. And the result demonstrates the superiority of the Rumi-GAN.

Clarity: The paper is well written. I enjoy reading it. Like the quote, “the art of knowing is knowing what to ignore” a lot. I also like figure 1, which introduce Rumi-GAN and ACGAN and GenPU succinctly and show the difference clearly.

Relation to Prior Work: Yes.

Reproducibility: Yes

Additional Feedback:

[Author Response · NeurIPS 2020]

We thank the reviewers for their valuable feedback. $\mathbf{R}m.n$ below refers to our response to comment $n$ from reviewer $m$.

**R1.1 Intuition behind Rumi-GANs:** As the title emphasizes, the motivation is to leverage *negative* class samples
to improve a GAN's capability of generating samples coming from the *positive* class. The proposed approach is
particularly useful when one is required to generate under-represented class samples (cf. Section 5). We showed that
presenting the GAN with information of the data manifold that it must avoid boosts its generation performance on the
desired manifold. We will add a few sentences in the Introduction of the final paper to further emphasize this point.

**R1.2 TAC-GAN and CCGAN:** A comparison with TAC-GAN is provided in Table 1 below and will be included in the
final paper. While Rumi-GANs emphasize the generative aspect, complementary learning frameworks such as CCGAN
(Xu *et al.*, arXiv, 2019) deal with *discriminative learning*. A comparative assessment will be included in the final paper.

**R1.3 Tabulation of FID and sensitivity to $\alpha^+$:** The converged FID scores for the experiments reported will be
included in a table (Table 1 below) in the final paper. The fraction of the generated positive class samples increases with
$\alpha^+$. Fig. 2(a) below shows FID scores evaluated with respect to the negative and positive classes versus $\alpha^+$.

**R1.4 Typo errors in the caption of Figure 1:** We have rectified the typographical error.

**R1.5 Novelty:** The Rumi framework is based on a new mathematical formulation aimed at leveraging negative samples
for boosting the generative capability for a desired target class. It is a *light-weight adaptation* and is applicable to any
existing GAN. Unlike discriminative learning approaches, the emphasis here is on the generative aspect (cf. **R1.2**).

**R2.1 Motivation and setting not clear:** Our objective is to obtain a better generator and thereby improve the
performance of GANs by leveraging undesirable samples to boost the generative capability for the desired class. The
objective function is appropriately formulated. We also demonstrated a concrete application to generating samples
belonging to under-represented classes in unbalanced datasets (cf. Section 5). Also refer to **R1.1** and **R1.5** above.

**R2.2 On splitting of data:** The designation of what constitutes a positive class or a negative class is a *design*
*specification*. Considering MNIST, we presented results highlighting the merit of the Rumi formulation for two
illustrative specifications of the positive and negative classes. In an unbalanced data application, the under-represented
class that is required to be modeled is the positive one and everything else is labelled negative (cf. Section 5).

**R2.3 Regularization of the generator loss:** Regularization functionals enforcing the integral and non-negativity
constraints are expressly needed in the Rumi framework. They apply to LSGAN as well (cf. Line 112 in the main
document and Section 1 of the supplementary). While the constraints may not be needed from an implementation
perspective, from a theoretical standpoint, they are necessary mathematical safeguards and have to be enforced explicitly.

**R3.1 Related works:** We'll discuss the suggested related methods in the Introduction section of the final paper.

**R3.2 Error in $\beta^+$:** We've fixed the bug in the main document. The supplementary (L32-L33) has the correct expression.

**R3.3 ACGAN on MNIST:** The ACGAN is indeed trained on the full dataset regardless of the positive/negative split.
However, we found that the ACGAN performance varies based on the split. Performing additional runs and averaging
the scores obtained was found to reduce that difference only marginally (cf. Table 1 below, ACGAN on MNIST).

**R3.4 High-resolution Images:** We resorted to experiments on low-resolution images due to Covid-19-related closure
of the compute facility at our university. We now have results on CelebA 128×128 images as well (cf. Fig. 1 below).

**R3.5 Domain-shift applications:** This is an interesting proposition. We deployed the Rumi-GAN formulation for the
experiment suggested by the reviewer (CelebA as the positive class and CIFAR-10 as the negative one). The FID curves
shown in Fig. 2(b) below indicate an improvement over the baseline demonstrating the strength of the formulation.

**R3.6 Averaging the results over classes:** Done. See Table 1 below (rows CelebA and CIFAR-10).

**R4.1 Benefit for real-world applications:** The Rumi formulation is promising for domain-shift applications (cf. **R3.5**)
and for handling data imbalance in medical image classification problems (similar to the one illustrated in Section 5).

**Figure 1:** Rumi-LSGAN results on high-resolution (128 × 128) CelebA images for *Bald* class and *Female* class.

**Figure 2:** (a) FID versus $\alpha^+$, and (b) FID improvement in a domain-shift application with the Rumi formulation.

| | | LSGAN | ACGAN | TAC-GAN | **Rumi-LSGAN** |
|---|---|---|---|---|---|
| **Balanced** | MNIST (Even) | 29.92 | 26.75 | 23.48 | **22.68** |
| | MNIST (Overlap) | 34.63 | 32.75 | 31.25 | **31.17** |
| **Unbalanced** | MNIST (Digit 5) | 148.55 | 127.91 | 121.3 | **118.93** |
| | CelebA (Averaged) | 326.25 | 243.95 | 213.6 | **169.34** |
| | CIFAR-10 (Averaged) | 231.02 | 455.06 | 275.43 | **217.15** |

**Table 1:** FID scores upon convergence for various GANs on both balanced and unbalanced datasets. Rumi-LSGAN has better FID scores.

[Meta-Review · NeurIPS 2020]

This paper presents Rumi-GAN, which aims to improve GAN by incorporating negative samples. Rumi-GAN can be easily applied to another GAN-based framework, which only requires to divide training data into the positive and negative ones. The authors compared with ACGAN, LSGAN, and the modified Rumi-LSGAN on several datasets. Reviewers agreed on the novelty after rebuttal. Also, some remaining issues should be addressed in the final version, e.g., more comparison with SOTA cGANs and more details on the FID scores.